# Multidimensional Evaluation for Detecting Salt Tolerance of Bread Wheat Genotypes Under Actual Saline Field Growing Conditions

**DOI:** 10.3390/plants9101324

**Published:** 2020-10-06

**Authors:** Elsayed Mansour, Ehab S. A. Moustafa, El-Sayed M. Desoky, Mohamed M. A. Ali, Mohamed A. T. Yasin, Ahmed Attia, Nasser Alsuhaibani, Muhammad Usman Tahir, Salah El-Hendawy

**Affiliations:** 1Agronomy Department, Faculty of Agriculture, Zagazig University, Zagazig 44519, Egypt; sayed_mansour@zu.edu.eg (E.M.); mohammed_ali@zu.edu.eg (M.M.A.A.); mayasein@zu.edu.eg (M.A.T.Y.); ahmedatia@zu.edu.eg (A.A.); 2Genetic Resources Department, Desert Research Center, Cairo 11753, Egypt; ehab.soudi@yahoo.com; 3Botany Department, Faculty of Agriculture, Zagazig University, Zagazig 44519, Egypt; sayed1981@zu.edu.eg; 4Department of Plant Production, College of Food and Agriculture Sciences, King Saud University, P.O. Box 2460, Riyadh 11451, Saudi Arabia; nsuhaib@ksu.edu.sa (N.A.); mtahir@ksu.edu.sa (M.U.T.); 5Department of Agronomy, Faculty of Agriculture, Suez Canal University, Ismailia 41522, Egypt

**Keywords:** agronomic parameters, enzymatic and nonenzymatic antioxidants, gas exchange, ion contents, principal component analysis

## Abstract

Field-based trials and genotype evaluation until yielding stage are two important steps in improving the salt tolerance of crop genotypes and identifying what parameters can be strong candidates for the better understanding of salt tolerance mechanisms in different genotypes. In this study, the salt tolerance of 18 bread wheat genotypes was evaluated under natural saline field conditions and at three saline irrigation levels (5.25, 8.35, and 11.12 dS m^−1^) extracted from wells. Multidimensional evaluation for salt tolerance of these genotypes was done using a set of agronomic and physio-biochemical attributes. Based on yield index under three salinity levels, the genotypes were classified into four groups ranging from salt-tolerant to salt-sensitive genotypes. The salt-tolerant genotypes exhibited values of total chlorophyll, gas exchange (net photosynthetic rate, transpiration rate, and stomatal conductance), water relation (relative water content and membrane stability index), nonenzymatic osmolytes (soluble sugar, free proline, and ascorbic acid), antioxidant enzyme activities (superoxide dismutase, catalase, and peroxidase), K^+^ content, and K^+^/Na^+^ ratio that were greater than those of salt-sensitive genotypes. Additionally, the salt-tolerant genotypes consistently exhibited good control of Na^+^ and Cl^−^ levels and maintained lower contents of malondialdehyde and electrolyte leakage under high salinity level, compared with the salt-sensitive genotypes. Several physio-biochemical parameters showed highly positive associations with grain yield and its components, whereas negative association was observed in other parameters. Accordingly, these physio-biochemical parameters can be used as individual or complementary screening criteria for evaluating salt tolerance and improvement of bread wheat genotypes under natural saline field conditions.

## 1. Introduction

Wheat is the most important field crop worldwide, and it is cultivated on more land area than any other field crop. It is the most important grain source for humans with an increasing demand all over the world. It is considered as the major source of starch and energy, and it supplies considerable amounts of protein, vitamins, dietary fiber, and phytochemicals [1,2]. The total area allocated for wheat farming in 2018 was 214.3 million hectares, producing 734.1 million tons [3]. Egypt is involved in these statistics, with 1.3 million hectares and 8.8 million tons in production. Moreover, Egypt is one of the top wheat importers, with approximately 10 million tons annually imported [3]. Importantly, the gap between production and consumption is increasing due to current and expected future population growth and climate change. This calls for expanding the production areas to marginal environments characterized with high salinity levels.

Salinity is one of the widespread abiotic stresses that adversely affect plant growth and productivity in several parts of the world, particularly in arid regions [4,5,6]. About 30% of the world irrigated lands that produce approximately one-third of the crops are affected by salinity [3,7]. Egypt suffers from severe salinity problems as about 35% of the cultivated lands are now within the range of the effects of salinity [8]. Furthermore, recent climate changes are characterized by rainfall reduction and temperature increase. These conditions may increase the negative effects of salinity on the agricultural sector of the arid and semiarid countries due to diminished salt leaching and increased evaporation demand of atmosphere [9].

Salinity stress negatively impacts growth performance and plant productivity through osmotic and specific ion toxic stresses due to excessive salt concentrations in the root zone and the excessive buildup of Na^+^ and Cl^−^ in the leaf blade, respectively. The former stress diminishes the plants’ ability to benefit from water, similar to water-deficit-induced stress [10,11]. However, the latter stress leads to an energy problem in plants due to a significant reduction in photosynthetic capacity, different metabolic functions, and cell elongation [12,13,14,15,16]. Moreover, salinity stress leads to nutrient imbalance by reducing uptake of essential elements, particularly K^+^, Mg^2+^, and Ca^2+^, which causes ion imbalance at cellular and tissue levels [15,17]. Finally, the different aspects of salinity stresses interact together to induce high reactive oxygen species (ROS) production in plant cells, including superoxide (O_2_^−^), hydroxyl radicals (OH), hydrogen peroxide (H_2_O_2_), and singlet oxygen (O_2_), [18,19,20]. These ROS damage photosynthetic pigments, nucleic acids, and membrane lipids, and inhibit protein synthesis and loss of enzymes activities, thereby decreasing CO_2_ pressure and photosynthetic efficiency [21,22,23]. Therefore, the devastating effects of salinity stress on different plant physiologic processes reflect on plant development, growth, and final productivity [14,24,25,26,27].

The plants mitigate ROS-induced damage by utilizing the complex antioxidant system [28,29]. This system includes both enzymatic antioxidants, such as superoxide dismutase (SOD), catalase (CAT), and peroxidase (POD); and non-enzymatic osmolytes, such as soluble sugar, free proline, and ascorbic acid [30,31]. Furthermore, plants also regulate ion mechanisms to tolerate salinity stress through selective preferential uptake and translocation of K^+^ ions over Na^+^ ions [32,33]. The primary object of this ion-regulating mechanism is to maintain a high K^+^/Na^+^ ratio and reduce toxic Na^+^ concentrations in the shoot cytoplasm, thus avoiding devastating salinity effects on several cell metabolic functions [14,34,35,36].

The close relationship between salinity tolerance mechanisms and the aforementioned physiologic parameters, such as Na^+^, Cl^−^, and K^+^ contents, photosynthetic pigments, gas exchange, and enzymatic and non-enzymatic antioxidants, especially under realistic salinity stress conditions, allows for using these parameters as useful and reliable screening criteria together with grain yield and related agronomic traits for investigating the salt tolerance of bread wheat genotypes. Assessing physiologic parameters along with agronomic traits has been previously documented to be an effective integrative approach to distinguish salt-tolerant wheat genotypes from salt-sensitive ones [14,36,37,38,39].

Cultivation of salt-tolerant genotypes is considered as an effective approach to cope with salinity conditions and obtain acceptable grain yield [40]. Indeed, wheat genotypes have different potentialities in adapting with salt stress and producing satisfactory grain yield under natural salinity stress conditions. Several researchers demonstrated considerable genetic variations in salinity tolerance in bread wheat using different experimental setups carried out under either controlled or simulated field conditions [14,41,42,43,44,45,46,47,48,49]. However, evaluating the salt tolerance of field crops should be confirmed until the yielding stage and, most importantly, should be performed under real field conditions, especially when the evaluation of salt tolerance happens in the advanced generation of breeding programs [14,50,51]. This is because under real field conditions, the plants are exposed to natural and realistic conditions, such as spatial and temporal heterogeneity of soil, low humidity, drought stress, and high differences of diurnal temperature, all of which occur simultaneously with salinity stress [52,53,54,55]. This contributes to the recognition of appropriate genotypes that can be grown in salt-stress-affected regions and identification of genotypes as source and guide for improving and salt tolerance of bread wheat in breeding programs.

The present study aimed to (i) assess the responses of various agronomical and physiologic parameters of 18 bread wheat genotypes with different genetic backgrounds to gradient levels of saline water under natural saline field conditions; (ii) classify the salt tolerance levels of the evaluated genotypes based on yield index; and (iii) investigate the inter-relationship between agronomical and physiologic parameters under natural saline field conditions.

## 2. Materials and Methods

### 2.1. Experimental Site and Water Source Description

Field experiment was conducted at the Ras-Sudr Experimental Station, Desert Research Center, Southwest Sinai, Egypt (29.6° N and 32.7° E) during the 2018–2019 and 2019–2020 growing seasons. The experimental site was arid with very low precipitation (an average annual rainfall of approximately 50 mm), and an average temperature of about 36 °C in the summer and 13.5 °C in winter. The average monthly temperature and accumulative precipitation during the two growing seasons as well as the long-term average of 38 years are shown in Appendix A.

Importantly, fresh water is scarce in the experimental site; therefore, the irrigation commonly depends on saline well water. Irrigation water was extracted from three wells with three different salinity levels: low (5.25 dS m^−1^), moderate (8.35 dS m^−1^), and high (11.12 dS m^−1^). The different chemical properties of the irrigation water from the three wells are shown in Appendix A. Surface irrigation was applied every week according to the standard practice for this region, with a cumulative amount of irrigation for the full growing season of approximately 500 mm ha^−1^.

The texture of the experimental soil is sandy loam throughout its profile, consisting of 86.95% sand, 8.75% silt, and 4.3% clay, with a pH of 8.15 and electrical conductivity (EC) of 7.74 dS m^−1^. The EC was determined using the soil water extraction method using suspensions with 1:1 air-dried soil-to-water ratio. The experimental soil also had large sand particles and very large pore spaces that facilitate rapid water leaching and prevent salt accumulation in the rooting zone. The different soil chemical properties are presented in Appendix A.

To mitigate the negative impacts of osmotic shock of the moderate and high salinity level treatments on germination and seedling growth, both treatments were first irrigated with low water salinity level (5.25 dS m^−1^) until full emergence; thereafter, moderate and high salinity level treatments were irrigated with saline water with an EC of 8.35 and 11.12 dS m^−1^, respectively. Furthermore, the soil moisture level for the three salinity treatments was kept near saturation during the early growth stage to avoid water deficit stress and therefore lessen salt stress effect on seed germination and stand establishment.

### 2.2. Plant Materials, Experimental Design, and Agronomic Practices

A total of 18 wheat genotypes were used in the present study (Appendix A). These genotypes represented an elite genetic material with varying yield potentials and, including 13 commercial cultivars, appear in the Egyptian recommended list, along with five CIMMYT advanced breeding lines.

The field experiment was conducted using a split-plot, randomized complete block design with three replicates in each growing season. The main plots correspond to the three irrigation salinity levels (5.25, 8.35, and 11.12 dS m^−1^), and the subplots consisted of the 18 wheat genotypes. Each main plot was 1.2 m wide and 54 m long. The subplots were randomly assigned within the main plots and were 1.2 m wide and 3 m long, with a 0.75-m alley between subplots and a 1-m alley between replicates. The seeds of each genotype were planted at a seeding rate of 350 seeds m^−2^ in six-row subplots. The seeds were drilled in rows spaced 20 cm apart on the third week of November in both growing seasons. Nitrogen, phosphorus, and potassium fertilizers were applied at rates of 180 N, 74 P_2_O_5_, and 115 K_2_O kg ha^−1^, respectively. The entire dose of phosphorus was applied before sowing as superphosphate (15.5% P_2_O_5_). Nitrogen fertilizer was applied as ammonium sulfate (21% N) in six split doses at 10-day intervals after sowing. Potassium fertilizer was applied as potassium sulfate (48% K_2_SO_4_) with the first two doses of the nitrogen fertilizer.

### 2.3. Measurements

#### 2.3.1. Physio-Biochemical Parameters

All physiologic parameters were measured on the sixth node leaf at 55 days from sowing. The youngest fully expanded leaves were excised and total chlorophyll concentration (Chl) was measured using pure acetone, according to the method previously described by Arnon [56]. The different parameters related to photosynthetic capacity (net photosynthesis rate, transpiration rate, and stomatal conductance) were measured in the leaves of three randomly selected plants from each subplot using a portable photosynthesis system (Li-6400XTR, Li-COR, Inc., Lincoln, NE, USA). The measurements were performed from 9:30 AM to 12:30 PM over two consecutive days on the second fully expanded leaf from the top of each plant. Relative water content (RWC), membrane stability index (MSI), and electrolyte leakage (EL) were measured using the fresh fully expanded leaves without the midrib, according to the methods previously outlined by Weatherley [57], Premachandra et al. [58], and Sullivan and Ross [59], respectively. Malondialdehyde (MDA) was assessed following the method previously described by Zhang and Qu [60]. Proline accumulation in leaves was measured as previously presented by Bates et al. [61]. The method previously described Irrigoyen et al. [62] was used to estimate the total soluble sugar content. Ascorbic acid (AsA) content was determined as previously outlined by Mukherjee and Choudhuri [63].

To investigate antioxidant enzyme activities, an extraction from 0.5 g of fresh leaves were taken, according to the method previously described by Mukherjee and Choudhuri [63]. The obtained extracts were frozen in liquid nitrogen and then powdered in a 100-mM phosphate buffer with a pH of 7.0. The homogenate was centrifuged at 15.000 × g at 4 °C for 10 min. Then, supernatants were preserved at 4 °C until being used for determining the activity of peroxidase (POD), catalase (CAT), and superoxide dismutase (SOD). SOD activity (EC1.15.1.1) was determined using the nitro blue tetrazolium (NBT) method previously presented by Giannopolitis and Ries [64], displaying its units as the required enzyme quantity to inhibit 50% NBT reduction as observed at 560 nm. CAT activity (EC1.11.1.6) was assessed following the method previously described by Aebi [65]. The reduction in absorbance at 240 nm as an outcome of H_2_O_2_ consumption revealed enzyme activity. POD activity (EC1.11.1.7) was determined as previously presented by Maechlay and Chance [66], and Klapheck et al. [67]. The rate of guaiacol oxidation in the presence of H_2_O_2_ read at 470 nm revealed enzyme activity.

To estimate Na^+^ and K^+^ concentrations, approximately 0.2 g of the oven-dried samples of leaves was digested using 96% H_2_SO_4_ in the presence of H_2_O_2_ and diluted with distilled water [68]. The concentrations of both ions were estimated using a flame photometer (ELEX 6361, Eppendorf AG, Hamburg, Germany) and, subsequently, the K^+^/Na^+^ ratios were calculated. Chloride concentration was determined using the simple water extraction method. About 0.1 g of the ground leaf sample was extracted in 10 mL of deionized water and shaken for 2 h, and then filtered. Then, Cl^−^ concentration was determined using a chloride analyzer (Model 926, Sherwood Scientific Ltd., Cambridge, UK).

#### 2.3.2. Agronomical Parameters

When the plants reached maturity (about 110 days after sowing), different agronomical parameters, namely plant height, spike length, number of spikes m^−2^, number of grains per spike, 1000-grain weight, and grain yield and biological yield per hectare, were evaluated. Plant height was measured from the ground to the spike top. The number of spikes in 0.5 m^2^ was recorded. The number of grains per spike was counted from 10 spikes randomly taken from each subplot.

An area of 2.4 m^2^ (four 3-m consecutive rows) were harvested, the spikes were threshed, and final grain yield was weighed and expressed as kg ha^−1^ based on the harvested plot area. The grain yield of individual genotype under each salinity level (Ys) and the average grain yield of all genotypes under the same salinity level (Ýs) were applied to calculate the yield index (YI) for individual genotype under each salinity level [69] using the following equation:(1)$$\mathrm{YI}=\frac{\mathrm{Ys}}{Ýs}$$

### 2.4. Statistical Analysis

Analysis of variance (ANOVA) was performed for the split-plot design in three replicates across the two growing seasons since yearly differences were insignificant. Combined ANOVA was performed to analyze the salinity and genotypic differences across the two growing seasons using the Shapiro-Wilk test and Bartlett’s test for the normality distribution of the residuals and homogeneity of variances, respectively. The combined analysis indicated homogenous variances across the two growing seasons for different parametric measurements, and therefore, the data of the two growing seasons were combined. Salinity level, genotype group, and their interaction were considered fixed effects. The growing season, replicate, and their interaction were considered random effects. The mean differences among salinity level, genotype group, and their interaction were compared using Fisher’s protected least significant difference test at a *p* ≤ 0.05 significance level.

Hierarchical cluster analysis was performed following the Ward method [70] to group the evaluated genotypes according to the level of salt tolerance (tolerant, moderately tolerant, moderately sensitive, and sensitive) based on their YI across the three salinity levels. The additive main effect and multiplicative interaction (AMMI) model [71] was performed to investigate genotypic stability patterns for grain yield across salinity levels during both growing seasons. Principal component analysis (PCA) was performed on the averages of the measured physiologic and agronomical parameters to determine their relationships. The R software version 3.6.1 [71] was used for this analysis.

## 3. Results

### 3.1. Grouping Genotypes Based in Their Salt Tolerance Level

The genotypes were classified using cluster analysis based on their grain yield and YI across three salinity levels into four distinct groups. The results in Table 1 show that the genotypes in group A, containing only one genotype (Gemiza-11), attained a higher value for grain yield (3975.0 kg ha^−1^) and YI (1.4); however, the opposite was true for the genotypes in group D, consisting of three genotypes (Gemiza-12, Misr-2, and Shandawel-1) and attaining lower values for grain yield (2089.3 kg ha^−1^) and YI (0.72). Therefore, the genotypes in group A and group D can be considered as salt-tolerant and salt-sensitive genotypes, respectively (Table 1). Group B and group C included six (Gemiza-9, Gemiza-10, Giza-171, Sids-14, Line-6083, and Line-6084) and eight (Giza-168, Gemiza-7, Sakha-94, Sids-12, Misr-1, Line-6052, Line-6078, and Line-1208) genotypes, respectively. The mean values of grain yield and YI of the genotypes in group B decreased by 14.6% and 16.9%, whereas the genotypes in group C decreased by 32.3% and 32.5%, respectively, compared with the mean values of the salt-tolerant genotype in group A. Additionally, the mean values of grain yield and YI of the genotypes in group B increased by 62.4% and 60.4%, whereas the genotypes in group C increased by 28.8% and 30.4%, respectively, compared with the mean values of the salt-sensitive genotypes in group D (Table 1). Therefore, the genotypes in group B and group C can be considered as moderately salt-tolerant and moderately salt-sensitive genotypes, respectively.

### 3.2. Physio-Biochemical Attributes

The combined ANOVA for Chl, gas exchange (net photosynthesis rate (*Pn*), transpiration rate (*E*), and stomatal conductance (*Gs*)), RWC, and MSI indicated that the effects of salinity level, genotype group, and their interaction on these parameters were highly significant (*p* ≤ 0.001) (Table 2). High salinity level (11.12 dS m^−1^) significantly diminished all above-mentioned physiologic attributes compared with moderate (8.35 dS m^−1^) and low (5.25 dS m^−1^) salinity levels. However, the highest values of these parameters were assigned for the genotype in group A. As compared with group A, the decreases in these parameters were more pronounced for groups D and C than for group B. The mean values of Chl, *Pn*, *E*, *Gs*, RWC, and MSI decreased by 6.6%, 3.1%, 3.7%, 4.7%, 4.4%, and 2.8% for the genotypes in group B, by 14.5%, 11.0%, 12.7%, 14.0%, 9.4%, and 9.3% for the genotypes in group C, and by 20.2%, 18.7%, 22.5%, 23.3%, 14.3%, and 18.6% for the genotypes in group D, respectively, compared with the mean values of the salt-tolerant genotype in group A (Table 2).

Lipid peroxidation, determined as the malondialdehyde content (MDA) and EL, was significantly increased by the increasing salinity level of the irrigation water (Table 3). The salt-tolerant genotypes in group A had the lowest values for both parameters; the opposite was true for the salt-sensitive genotypes in group D. Furthermore, the contents of non-enzymatic osmolytes (soluble sugar, free proline, and AsA) were significantly increased with the increasing salinity level of irrigation water. However, the genotypes in group A displayed the highest values of these osmoprotectants, followed by groups B and C, whereas group D had the lowest values. Moreover, the activities of enzymatic antioxidants, such as SOD, CAT, and POD, were significantly superior under high salinity level than in low and moderate levels. On another note, group A presented the highest values, followed by groups B and C, whereas group D had the lowest values (Table 3).

Based on the ANOVA analysis, the effects of salinity level, genotype group, and their interaction on leaf ion contents (Cl^−^, Na^+^, and K^+^), as well as K^+^/Na^+^ ratio, were highly significant (*p* ≤ 0.001) (Table 4). Generally, as the salinity level of the irrigation water increased, the contents of toxic ions (Cl^−^ and Na^+^) also increased, whereas the content of essential ions and their ratios with Na^+^ (K^+^ and K^+^/Na^+^ ratio) increased. However, the genotypes in groups A and B always demonstrated good control of Cl^−^ and Na^+^ accumulation and maintained higher K^+^ content and K^+^/Na^+^ ratio than the genotypes in group D (Table 4).

### 3.3. Agronomical Attributes

Generally, as the salinity levels of irrigation water increased, the values of grain yield and all related agronomic traits were significantly decreased (Table 5). The evaluated genotypes presented considerable variation for grain yield and related traits under salinity levels varied from 1246 to 5392 kg ha^−1^ (Appendix A). High salinity level reduced grain yield by 44.9% compared with low level. However, the genotype in group A exhibited significant increase in grain yield (47.4%), followed by group B (38.3%), and group C (22.4%), compared with the genotypes in group D (Table 5).

### 3.4. Additive Main Effect and Multiplicative Interaction Model (AMMI)

The AMMI model was applied to investigate the stability of the evaluated wheat genotypes across the three salinity levels of irrigation water during both growing seasons (in a total of six environments). The first two PCAs explained 79.9% and 10.2% of the genotype-by-environment interaction (GEI) effect, respectively (Figure 1). The genotypes located nearby the origin and exhibited lower values of the PCAs close to zero proved low contribution to GEI and are more stable across the three salinity levels, compared with those far from the origin. The results revealed that Gemiza-11 (salt-tolerant genotype); Line-6083 (moderately salt-tolerant genotype); and Gemiza-7, Sids-12, Line-1208, and Line-6078 (moderately salt-sensitive genotypes) showed the least variable values on the AMMI model and had good stability. On another note, Giza-171, Gemiza-9, Line-6084, and Gemiza-10 (moderately salt-tolerant genotypes); Giza-168 and Misr-1 (moderately salt-sensitive genotypes); and Misr-2 and Gemiza-12 (salt-sensitive genotypes) were located further from origin. Additionally, the environments that presented three salinity levels in two growing seasons were located in diverse sites, revealing their high diversity and their large contribution to GEI.

### 3.5. Inter-Relationship Between All Measured Parameters

The association between all agronomical and physiologic parameters under salinity stress conditions was estimated through PCA. The first two PCAs were used to construct an informative plot system based on the evaluated wheat genotypes. The first two PCAs presented about 87.31% of variability, with the first and second PCA explaining 79.24% and 8.07% of the total variations between all parameters, respectively (Figure 2). Generally, the parameters that were represented by adjacent or parallel vectors to each other indicate strong positive association between themselves. However, the vectors toward the sides expressed weak relationships and those placed approximately opposite (at 180°) displayed highly negative associations. All the measured parameters in the present study can be divided into three groups. The first group contained all agronomical parameters (plant height, spike length, number of spikes per square meter, number of grains per spike, 1000-grain weight, and grain yield per hectare), as well as including eight out of 18 physio-biochemical parameters (Chl, *Pn*, *E*, *Gs*, RWC, MSI, K^+^ content, and K^+^/Na^+^ ratio) with an acute angle between the vectors of these parameters. The second group consisted of the non-enzymatic-enzymatic osmolyte measurements (soluble sugar, free proline content, and AsA), as well the activities of three enzymatic antioxidants (SOD, CAT, and POD), with an angle less than 90° between the vectors of these parameters and those of the agronomic parameters, as well as with those of the aforementioned eight physio-biochemical parameters. The third group included MDA and toxic ion (EL, Cl^−^, and Na^+^) contents with a straight angle between the vectors of these parameters and those of all the aforementioned parameters. A strong positive association was detected among the traits included in the same group. Furthermore, a positive association was noticed between the second and third groups. On another note, a negative association was observed between the first and third groups (Figure 2).

## 4. Discussion

Large agricultural areas in arid and semiarid regions worldwide are affected by salinity problems that are projected to worsen due to abrupt climate changes [9,72]. Although effective drainage schemes and leaching enhancements can leach the salt away from the root zone, this strategy is still prohibitively costly and requires abundant fresh water, making it unfeasible on a large scale [14,73]. In a case similar to our study, salinity exists in both soil and irrigation water. Therefore, developing new genotypes that can produce sufficient and stabile yield under such conditions by enhancing their salt tolerance is one of the most effective and feasible ways to sustain crop production under salinity stress conditions. Unfortunately, the success in enhancing genotypic salt tolerance is very limited. One factor for this limitation is that the majority of salinity studies that evaluate genotypic salt tolerance were carried out under tightly controlled conditions, and these studies are focused on evaluating genotypic salt tolerance at early growth stages [14,36,45,46,47,49]. More importantly, the relative studies seldom use physio-biochemical attributes as evaluation criteria for genotypic salt tolerance, although these attributes are functions of various physiologic processes and reflect the response of these processes to salt stress at the canopy, organ, tissue, and cellular levels [39,74,75]. Therefore, it is vital to evaluate the response of wheat germplasm to salt stress until the yielding stage, because the response of wheat genotypes to salinity stress varies from one developmental stage to another [39,51,73]. Most importantly, it is necessary to evaluate wheat genotypes under natural salt-stressed field conditions in order to expose plants to fluctuating salt and water contents that concurrently take place in the root zone with high variability in macro- and micro-environmental conditions that surround the canopy of plants [14,51,75]. In the present study, the salt tolerance of 18 wheat genotypes with various genetic backgrounds was evaluated under field conditions by growing them in naturally salt-affected soil and irrigated with water that was extracted from three wells, having different salinity levels ranging from low (5.25 dS m^−1^) to high (11.12 dS m^−1^) (Appendix A). Additionally, genotypic salt tolerance was evaluated until the yielding stage by monitoring the changes that take place in several physio-biochemical and agronomic parameters, which shows the response to salt stress at various organizational levels in the plant (whole canopy, tissue, and cellular levels).

The results revealed highly significant differences among salinity levels and genotypes for all the investigated physio-biochemical and agronomic parameters (Table 1, Table 2, Table 3, Table 4 and Table 5). These findings demonstrate the presence of considerable variability in the genotypes with different responses to different salinity levels. Subsequently, the genotypes were classified into four groups according to their level of salt tolerance, which was determined according to seed yield and YI values across three salinity levels (Table 1). Similarly, El-Hendawy et al. [14], El-Hendawy et al. [37], Hasan et al. [38], Oyiga et al. [39], Houshmand et al. [76], and Shafi et al. [77] evaluated the physiologic and agronomic parameters of different wheat genotypes under salinity stress field conditions and observed highly significant variations among the evaluated genotypes in terms of their responses to salinity stress, and these parameters were effective to distinguish the salt-tolerant from the salt-sensitive genotypes.

Indeed, excessive salt concentrations in the root zone impedes the regulation of the net uptake of essential ions or harmful ions, and an imbalance in either of them causes a reductions in leaf chlorophyll concentration and photosynthetic efficiency [78,79,80]. Furthermore, overproduction of ROS, including H_2_O_2_, O_2_^−^, and OH as a result of salinity stress contributes to corrupted chlorophyll levels and pigment degradation, which is considered as an oxidative harm marker [20,81,82,83]. Additionally, salinity stress decreases the root hydraulic conductivity and reduces the water reuptake requirement from the soil, which in turn causes a significant reduction in leaf water content and leads to decreased transpiration rate and photosynthesis efficiency [14,84,85,86]. In the present study, the genotypes irrigated with moderate (8.35 dS m^−1^) and high (11.12 dS m^−1^) salinity level induced a significant reduction in the parameters related to gas exchange efficiency (Chl contents, *Pn*, *E*, and *Gs*) and water relation (RWC and MSI), compared with low salinity level irrigation (5.25 dS m^−1^) (Table 2). However, the salt-tolerant and moderately salt-tolerant genotypes in groups A and B, respectively, displayed improved performance in these parameters, compared with the moderate salt-sensitive and salt-sensitive genotypes in groups C and D, respectively.

The plants utilized a complex antioxidant defense system to mitigate salt-stress-induced damage [29,87]. This system includes non-enzymatic solutes, such as soluble sugar, free proline, and AsA, together with enzymatic antioxidants, such as SOD, CAT, and POD [30,31]. In the present study, the aforementioned enzymatic and non-enzymatic antioxidants were higher in either salt-tolerant (group A) or moderately salt-tolerant (group B) wheat genotypes (Table 3). These results indicate that the ability of the genotypes to utilize their antioxidant system plays an important role in limiting the damage caused by salinity stress [88,89]. Many plants accumulate non-enzymatic solutes as a protective and nontoxic osmolytes under salinity stress [26]. The non-enzymatic osmolytes significantly contribute to salt tolerance through its dynamic role in cellular osmotic adjustment, cell membrane protection, and mitigation of toxic ion effects under salinity stress [90,91]. The accumulation of these osmoprotectants in plant cells is an important mechanism for preserving water that is needed by other physio-biochemical processes to safeguard against the negative impacts of salinity stress. Moreover, their accumulation preserves the balance between the osmotic capacity of the cytosol and that of the vacuole under salinity stress [31,92]. Furthermore, the different enzymatic antioxidants play an important role in controlling ROS generation during salt stress by reducing O_2_ radicals and H_2_O_2_ concentration, and responding directly to OH^−^ levels [93,94,95,96]. POD and CAT constitute the primary cellular H_2_O_2_-scavenging systems by converting them into water and oxygen. In addition, SOD is considered as the most effective intracellular enzymatic antioxidant, because it provides the first line of defense against ROS toxicity [97,98,99,100]. Therefore, the different antioxidant enzymes support the salt-tolerant genotypes by avoiding the oxidative damage caused by salinity stress [93,101,102,103].

Malondialdehyde (MDA) is one of the indicators of lipid peroxidation and is appropriate for evaluating the plants’ tolerance or sensitivity to salt stress [104,105]. Likewise, EL is a useful physiologic parameter for distinguishing between salt-tolerant and salt-sensitive genotypes. Both components diminish cell membrane integrity, cellular water content, and metabolic functions under salinity stress conditions [89,106]. The results of the present study indicate that as the salinity level of the irrigation water increased, the MDA and EL contents also significantly increased (Table 3). However, salt-tolerant genotypes in groups A and B maintained lower MDA and EL contents, compared with the salt-sensitive genotypes in groups C and D (Table 3). The lower MDA and EL levels in the salt-tolerant genotypes reflected low oxidative damage and lipid peroxidation levels, thereby preserving the structure and stability of plasma membranes under salt stress.

In the present study, salinity stress elevated Na^+^ and Cl^−^ concentrations, in association with decreased leaf K^+^ concentration and K^+^/Na^+^ ratio (Table 4). In general, high Na^+^ and Cl^−^ concentrations in the root zone induced specific ion toxicity and/or nutritional imbalance due to the intense competition between Na^+^ and other essential ions, such as K^+^, Ca^2+^, and Mg^2+^, at the uptake site of ions in roots [14,26,27,39,107]. For instance, the uptake of K^+^ is adversely affected by high external Na^+^ concentration due to their similarities in terms of chemical properties [108]. Additionally, Na^+^ and Cl^−^ accumulation in plant cells lead to decreased Mg^2+^ synthesis, subsequently resulting in chlorophyll degradation, as evidenced by the burnt appearance of the edges of the leaves [109]. Therefore, the ability of genotypes to minimize Na^+^ and Cl^−^ accumulation, especially in the metabolically active areas of cells, coincides with their higher affinity for K^+^, Mg^2+^, and Ca^2+^ over Na^+^ in terms of ion uptake, which appeared to significantly improve their salt tolerance. In the present study, ion analysis revealed that the mechanism of salt tolerance is strongly associated with the ability to restrict the uptake of toxic ions (Na^+^ and Cl^−^), which also coincides with the high affinity for K^+^ over Na^+^ uptake. The salt-tolerant genotypes (genotypes classified in groups A and B) possessed lower Na^+^ and Cl^−^ concentrations, and higher accumulated K^+^ with a higher K^+^/Na^+^ ratio, compared with the salt-sensitive genotypes in groups C and D (Table 4). These results reveal that selective uptake of K^+^ as opposed to Na^+^ is considered to be one of the pivotal physiologic mechanisms contributing to the salt tolerance of these wheat genotypes under natural salinity field conditions. Certainly, under salinity stress, K^+^ ions play a distinct role in osmotic adjustment without the energy cost incurred during compatible organic substances synthesis, which is important to induce high water retention in plant cells, thereby promoting survival [14,75,110], and in maintaining plant leaf turgor, which is important in regulating stomatal opening. Indeed, stomatal regulation is a substantial process that affects the plant’s photosynthetic rate under salinity stress conditions. Therefore, the abundant presence of K^+^ within the cytosol enhanced the tolerance mechanisms of wheat genotypes under salinity stress [14,86,111,112].

Grain yield and its components are the final outputs of several physiologic processes that happen at different developmental stages [113]. In the present study, the grain yield and its related components gradually decreased with increasing salinity levels of irrigation water, showing different responses for grain yield and its components between the evaluated genotypes under salinity stress conditions (Table 5). Obviously, the considerable reduction in grain yield was primarily from the great reduction in their contributing components, namely plant height, spike length, number of spikes per m^2^, number of grains per spike, and 1000-grain weight. These components were gradually decreased with increasing salinity levels. Compared with low salinity level of irrigation water (5.25 dS m^−1^), the percentage reduction in these components reached 7.7%, 9.6%, 11.3%, 12.4%, and 10.1% in the moderate salinity level (8.35 dS m^−1^), and 18.4%, 18.5%, 20.2%, 24.5%, and 22.4% in the high salinity level (11.12 dS m^−1^), respectively (Table 5). These findings indicated that these different agronomic parameters were sensitive and adversely affected by salinity stress. The sensitivity of these parameters indicated their importance in developing and evaluating the salt tolerance of the wheat genotypes. In respect of genotypic performance, the genotypes in groups A and B displayed higher agronomic performance compared with groups C and D. The agronomic performance of these genotypes reflected their superior efficiency in net photosynthesis rate, stomatal conductance, RWC, MSI, enzymatic and non-enzymatic antioxidants, K^+^ content, and K^+^/Na^+^ ratio. Likewise, the salt-tolerant genotypes maintained lower Na^+^ and Cl^−^ contents, malondialdehyde levels, and EL values. Similarly, El-Hendawy et al. [37], Hasan et al. [38], Oyiga et al. [39], Houshmand et al. [76], and Shafi et al. [77] observed significant differences in agronomic parameters among wheat genotypes under salinity stress.

Using multiple parameters for identify genotypic salt tolerance provided useful information and increased the accuracy in classifying the genotypes based on salt tolerance. Multiple parameters were previously applied by El-Hendawy et al. [73], Zeng et al. [114], and Hammami et al. [115] in distinguishing salt-tolerant genotypes in different cereal crops such as rice, wheat, and barley, respectively. Therefore, in the present study, cluster analysis based on YI was effective in differentiating the salt-tolerant and salt-sensitive genotypes. The cluster analysis classified the salt tolerance of the evaluated genotypes into four groups (A–D) ranging from salt-tolerant to salt-sensitive genotypes. The Gemiza-11, Gemiza-9, Giza-171, Gemiza-10, Line-6084, Sids-14, and Line-6083 genotypes were identified as salt-tolerant and moderately salt-tolerant genotypes across three salinity stress levels (Table 1). However, Gemiza-11 and Line-6083 showed the least variable values on the PCs of AMMI, thereby exhibiting good stability through the three salinity levels [116]. These salt-tolerant and stable genotypes can be important sources of salt-tolerant lines for exploitation in wheat breeding programs.

Investigating the inter-relationships between agronomic and physio-biochemical parameters can provide useful information for screening the most suitable wheat genotypes under salinity conditions. The first two PCAs were used to construct informative biplots based on averages of the genotypes. Generally, the acute angle between the vectors of the evaluated parameters indicates that they were strongly and positively associated with each other. This suggested that most of the physio-biochemical parameters measured in this study can be used as good indicators of grain yield and its related yield components, and as individual or interchangeable screening criteria for salt tolerance. Conversely, the straight angle between the vectors indicated that these parameters were negatively associated with each other. The straight angle between toxic ions (Na^+^ and Cl^−^) and the agronomic and most of the physio-biochemical parameters suggests that the exclusion or restricted influx of both ions is still an effective screening criterion for evaluating the salt tolerance of wheat genotypes and an important mechanism in salinity tolerance. Additionally, Na^+^ concentration and its ratio with K^+^ can be used as an alternative screening criterion for salt tolerance. Moreover, the different nonenzymatic osmolytes and enzymatic antioxidants providing an acute angle (less than 90°) between their vectors and those of grain yield and its components. This indicates that these parameters are also important components in evaluating the salt tolerance of wheat genotypes under normal salinity field conditions, because they comprise an additional constituent in the genotypic salt tolerance mechanism against oxidative stress. Therefore, these parameters can be used as individual or complementary screening criteria for evaluating the salt tolerance of wheat genotypes under field conditions. These findings are consistent with previous studies that indicate several important physio-biochemical parameters that are useful and effective screening criteria for evaluating salt tolerance of wheat genotypes and as good indicators for grain yield and its related components under salinity stress [14,36,49].

## 5. Conclusions

This study has given further weight to using several agronomic and physio-biochemical parameters as reliable screening criteria for selecting and improving the salt tolerance of wheat genotypes, especially since these parameters were tested under realistic field conditions, and the genotypes were evaluated until the yielding stage. The 18 wheat genotypes had different genetic backgrounds and different responses to salinity stress. The salt tolerance of these genotypes was classified into four groups based on their YI under three salinity levels simultaneously. Overall, increasing salinity level significantly decreased the values of the different parameters related to photosynthesis efficiency, water relation, essential ion uptake, and final grain yield and its components. In contrast, the different measurements related to MDA content, EL value, non-enzymatic osmolyte concentration, and antioxidant enzyme activity, as well as toxic ion uptake, significantly increased with the increasing salinity levels. The salt-tolerant and moderately salt-tolerant genotypes showed evidence of possessing a more efficient mechanism against salt stress by protecting themselves from ion toxicity and osmotic injury effects; and maintained higher contents of K^+^, photosynthetic pigments, and non-enzymatic osmolytes, higher gas exchange efficiency, and antioxidant enzyme activity under salinity stress than those of the salt-sensitive genotypes. Therefore, these genotypes and these parameters can be considered as potential candidates for better understanding of the protective mechanisms in wheat under realistic saline field conditions and their application in breeding programs for efficiently improving salt-tolerant genotypes. The PCA provided a comprehensive picture of the inter-relationships between these parameters, and it also identified which parameters can be individually or interchangeably used as the screening criteria for evaluating the salt tolerance of wheat genotypes under real saline field conditions.

## Figures and Tables

**Figure 1 plants-09-01324-f001:**
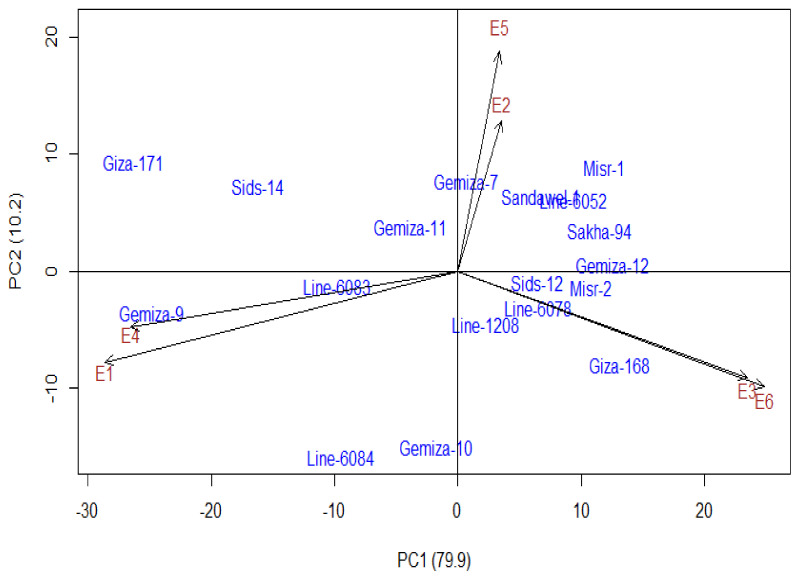
Additive main effect and multiplicative interaction (AMMI) biplot for grain yield of 18 wheat genotypes evaluated in six environments. E1 and E4 are low salinity level (5.25 dS m^−1^ in the two growing seasons), E2 and E5 are moderate salinity level (8.35 dS m^−1^ in the two growing seasons), and E3 and E6 are high salinity level (11.12 dS m^−1^ in the two growing seasons).

**Figure 2 plants-09-01324-f002:**
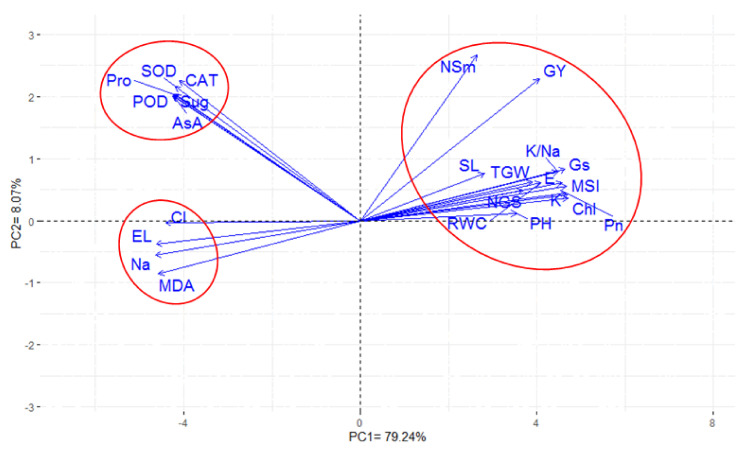
Biplot of the principal component analysis for the first two principle components of agronomic and physiological traits at three salinity levels. PH is plant height, SL is spike length, NSm is number of spikes per square meter, NGS is number of grains per spike, TGW is 1000-grain weight, GY is grain yield, Chl is total chlorophyll, Pn is net photosynthetic rate, E transpiration rate, Gs is stomatal conductance, RWC is relative water content, MSI is membrane stability index, Sug is soluble sugars content, Pro is free proline content, AsA is ascorbic acid, SOD is superoxide dismutase, CAT catalase, POD peroxidase, MDA is malondialdehyde, EL is electrolyte leakage, Cl is chlorine content, Na is sodium content, K is potassium content, and K/Na is K/Na ratio.

**Table 1 plants-09-01324-t001:** Grain yield and YI of 18 wheat genotypes under the three salinity levels of S1 (5.25 dS m^−1^), S2 (8.35 dS m^−1^), and S3 (11.12 dS m^−1^). The value averaged over the two growing seasons. A, B, C, and D indicate salt-tolerant, moderately salt-tolerant, moderately salt-sensitive, and salt-sensitive genotypes, respectively.

Genotypes	Grain Yield (kg ha^−1^)				Yield Index (YI)				Cluster Group
	S1	S2	S3	Mean	S1	S2	S3	Mean	
Gemiza-11	4960	3973	2992	3975	1.30	1.40	1.47	1.39	A
Gemiza-9	5392	3549	2359	3767	1.42	1.25	1.16	1.27	B
Gemiza-10	4263	2849	2460	3191	1.12	1.00	1.20	1.11	B
Giza-171	5209	3680	2050	3646	1.37	1.29	1.00	1.22	B
Sids-14	4602	3339	1976	3306	1.21	1.17	0.97	1.12	B
Line-6084	4718	3080	2525	3441	1.24	1.08	1.24	1.19	B
Line-6083	4210	2892	1937	3013	1.11	1.02	0.95	1.02	B
Giza-168	3182	2442	2130	2585	0.84	0.86	1.04	0.91	C
Gemiza-7	3560	2768	1809	2712	0.94	0.97	0.89	0.93	C
Sakha-94	3185	2681	2007	2624	0.84	0.94	0.98	0.92	C
Sids-12	3613	2854	2193	2887	0.95	1.00	1.07	1.01	C
Misr-1	3254	2840	2075	2723	0.86	1.00	1.02	0.96	C
Line-6052	3185	2677	1884	2582	0.84	0.94	0.92	0.90	C
Line-6078	3360	2551	1947	2619	0.88	0.90	0.95	0.91	C
Line-1208	3669	2690	2041	2800	0.96	0.95	1.00	0.97	C
Gemiza-12	2786	2225	1668	2226	0.73	0.78	0.82	0.78	D
Shandawel-1	2680	2097	1246	2008	0.70	0.74	0.61	0.68	D
Misr-2	2659	1983	1462	2034	0.70	0.70	0.72	0.70	D

S1 is high salinity level (5.25 dS m^−1^), S2 is moderate salinity level (8.35 dS m^−1^), and S3 is low salinity level (11.12 dS m^−1^).

**Table 2 plants-09-01324-t002:** Impact of different salinity levels on total chlorophyll concentration, net photosynthetic rate (*Pn*), transpiration rate (*E*), stomatal conductance (*Gs*), relative water content (RWC), and membrane stability index (MSI) of four genotype groups classified according to salinity tolerance under three salinity levels of S1 (5.25 dS m^−1^), S2 (8.35 dS m^−1^), and S3 (11.12 dS m^−1^).

Genotypes	Total Chlorophyll (mg g^−1^ Fresh Weight)				*Pn* (µmol CO_2_ m^−2^ s^−1^)				*E* (mmol H_2_O m^−2^ s^−1^)			
	S1	S2	S3	Mean	S1	S2	S3	Mean	S1	S2	S3	Mean
Group A *n* = 1	2.96	2.22	1.66	**2.28 ^a^†**	12.72	10.14	6.89	**9.92 ^a^**	7.05	4.73	3.55	**5.11 ^a^**
Group B *n* = 6	2.92	2.09	1.39	**2.13 ^b^**	12.50	9.91	6.42	**9.61 ^b^**	7.01	4.52	3.22	**4.92 ^b^**
Group C *n* = 8	2.67	1.90	1.30	**1.95 ^c^**	11.24	9.18	6.07	**8.83 ^c^**	6.16	4.17	3.05	**4.46 ^c^**
Group D *n* = 3	2.50	1.81	1.15	**1.82 ^d^**	10.60	8.39	5.24	**8.07 ^d^**	5.37	3.72	2.80	**3.96 ^d^**
Mean	**2.76 ^A^**	**2.00 ^B^**	**1.37 ^C^**		**11.77 ^A^**	**9.40 ^B^**	**6.16 ^C^**		**6.40 ^A^**	**4.29 ^B^**	**3.15 ^C^**	
ANOVA	**df**	***p*-value**		**LSD**		***p*-value**		**LSD**		***p*-value**		**LSD**
Salinity (S)	2	< 0.001		0.011		< 0.001		0.07		< 0.001		0.15
Group (G)	3	< 0.001		0.029		< 0.001		0.11		< 0.001		0.16
S × G	6	< 0.001		0.047		< 0.001		0.20		0.003		0.29
Genotypes	***Gs* (mmol H_2_O m^−2^ s^−1^)**				**RWC (%)**				**MSI (%)**			
	**S1**	**S2**	**S3**	**Mean**	**S1**	**S2**	**S3**	**Mean**	**S1**	**S2**	**S3**	**Mean**
Group A *n* = 1	0.59	0.40	0.30	**0.43 ^a^**	65.81	51.56	42.04	**53.13 ^a^**	60.78	49.01	36.22	**48.67 ^a^**
Group B *n* = 6	0.56	0.39	0.27	**0.41 ^b^**	63.15	50.44	38.71	**50.77 ^b^**	59.96	48.07	33.90	**47.31 ^b^**
Group C *n* = 8	0.49	0.36	0.26	**0.37 ^c^**	56.99	47.89	39.48	**48.12 ^c^**	55.45	44.35	32.60	**44.13 ^c^**
Group D *n* = 3	0.44	0.33	0.21	**0.33 ^d^**	53.93	46.39	36.36	**45.56 ^d^**	51.18	40.71	26.90	**39.60 ^d^**
Mean	**0.52 ^A^**	**0.37 ^B^**	**0.26 ^C^**		**59.97 ^A^**	**49.07 ^B^**	**39.14 ^C^**		**56.84 ^A^**	**45.54 ^B^**	**32.40 ^C^**	
ANOVA	**df**	***p*-value**		**LSD**		***p*-value**		**LSD**		***p*-value**		**LSD**
Salinity (S)	2	< 0.001		0.003		< 0.001		1.71		< 0.001		0.16
Group (G)	3	< 0.001		0.006		< 0.001		1.42		< 0.001		0.32
S × G	6	< 0.001		0.010		0.002		2.77		< 0.001		0.55

† Means in each column or row followed by the same lowercase or uppercase letter, respectively, are not significantly different at *p* ≤ 0.05 according to Fisher’s protected LSD test.

**Table 3 plants-09-01324-t003:** Impact of different salinity levels on malondialdehyde (MDA), electrolyte leakage (EL), total soluble sugars content, free proline content, ascorbic acid (AsA), superoxide dismutase (SOD), catalase (CAT), and peroxidase (POD) of four genotype groups classified according to salinity tolerance under three salinity levels of S1 (5.25 dS m^−1^), S2 (8.35 dS m^−1^), and S3 (11.12 dS m^−1^).

Genotypes	MDA (nmol g^−1^ Fresh Weight)				EL (%)				Soluble Sugars (mg g^−1^ Dry Weight)				Free Proline (µg g^−1^ Dry Weight)			
	S1	S2	S3	Mean	S1	S2	S3	Mean	S1	S2	S3	Mean	S1	S2	S3	Mean
Group A *n* = 1	49.33	60.18	69.00	**59.50 ^d^†**	2.80	6.60	10.46	**6.62 ^d^**	25.95	40.53	56.80	**41.09 ^a^**	127.7	192.5	242.8	**187.7 ^a^**
Group B *n* = 6	50.07	61.55	72.64	**61.42 ^c^**	3.05	6.93	12.51	**7.49 ^c^**	25.89	39.57	53.26	**39.57 ^b^**	123.5	183.7	230.5	**179.2 ^b^**
Group C *n* = 8	53.65	64.26	74.94	**64.28 ^b^**	4.20	7.63	13.12	**8.32 ^b^**	24.11	35.23	50.03	**36.46 ^c^**	98.6	164.1	223.7	**162.2 ^c^**
Group D *n* = 3	57.61	67.92	82.56	**69.36 ^a^**	5.36	8.69	15.07	**9.71 ^a^**	21.79	30.62	44.66	**32.36 ^d^**	84.7	154.7	207.5	**148.9 ^d^**
Mean	**52.67 ^C^**	**63.48 ^B^**	**74.78 ^A^**		**3.85 ^C^**	**7.46 ^B^**	**12.79 ^A^**		**24.43 ^C^**	**36.48 ^B^**	**51.19 ^A^**		**108.6 ^C^**	**173.8 ^B^**	**226.1 ^A^**	
ANOVA	**df**	***p*-value**		**LSD**		***p*-value**		**LSD**		***p*-value**		**LSD**		***p*-value**		**LSD**
Salinity (S)	2	< 0.001		0.57		< 0.001		0.20		< 0.001		1.44		< 0.001		1.64
Group (G)	3	< 0.001		0.62		< 0.001		0.28		< 0.001		1.29		< 0.001		1.82
S × G	6	< 0.001		1.04		< 0.001		0.49		< 0.001		2.93		< 0.001		2.31
Genotypes	**AsA (µmol g^−1^ dry weight)**				**SOD (U µg^−1^ protein)**				**CAT (U mg^−1^ min^−1^)**				**POD (µg g^−1^ fresh weight min^−1^)**			
	**S1**	**S2**	**S3**	**Mean**	**S1**	**S2**	**S3**	**Mean**	**S1**	**S2**	**S3**	**Mean**	**S1**	**S2**	**S3**	**Mean**
Group A *n* = 1	1.74	2.53	3.03	**2.43 ^a^**	52.41	69.93	87.46	**69.93 ^a^**	0.50	0.70	0.96	**0.72 ^a^**	90.33	131.57	177.93	**133.28 ^a^**
Group B *n* = 6	1.72	2.46	2.86	**2.35 ^b^**	51.92	67.42	82.62	**67.32 ^b^**	0.50	0.65	0.85	**0.67 ^b^**	88.94	126.67	167.34	**127.65 ^b^**
Group C *n* = 8	1.61	2.25	2.77	**2.21 ^c^**	47.43	63.49	79.78	**63.57 ^c^**	0.46	0.60	0.81	**0.62 ^c^**	82.22	113.58	161.78	**119.19 ^c^**
Group D *n* = 3	1.53	2.08	2.56	**2.06 ^d^**	42.27	58.10	73.71	**58.03 ^d^**	0.42	0.57	0.73	**0.57 ^d^**	77.53	102.46	143.91	**107.97 ^d^**
Mean	**1.65 ^C^**	**2.33 ^B^**	**2.80 ^A^**		**48.51 ^C^**	**64.74 ^B^**	**80.89 ^A^**		**0.47 ^C^**	**0.63 ^B^**	**0.84 ^A^**		**84.76 ^C^**	**118.57 ^B^**	**162.74 ^A^**	
ANOVA	**df**	***p*-value**		**LSD**		***p*-value**		**LSD**		***p*-value**		**LSD**		***p*-value**		**LSD**
Salinity (S)	2	< 0.001		0.009		< 0.001		0.27		< 0.001		0.009		< 0.001		0.94
Group (G)	3	< 0.001		0.013		< 0.001		0.56		< 0.001		0.012		< 0.001		1.41
S × G	6	< 0.001		0.023		< 0.001		0.95		< 0.001		0.015		< 0.001		2.45

† Means in each column or row followed by the same lowercase or uppercase letter, respectively, are not significantly different at *p* ≤ 0.05 according to Fisher’s protected LSD test.

**Table 4 plants-09-01324-t004:** Impact of different salinity levels on ion contents in leaves (Cl^−^, Na^+^ and K^+^) and K^+^/Na^+^ ratio of four genotype groups classified according to salinity tolerance under three salinity levels of S1 (5.25 dS m^−1^), S2 (8.35 dS m^−1^), and S3 (11.12 dS m^−1^).

Genotypes	Cl^−^ (%)				Na^+^ (%)			
	S1	S2	S3	Mean	S1	S2	S3	Mean
Group A *n* = 1	1.55	2.05	3.30	**2.30 ^c^†**	4.35	8.21	11.15	**7.90 ^d^**
Group B *n* = 6	1.57	2.21	3.18	**2.32 ^c^**	4.56	8.77	12.18	**8.50 ^c^**
Group C *n* = 8	1.75	2.61	3.51	**2.62 ^b^**	5.84	9.76	12.87	**9.49 ^b^**
Group D *n* = 3	1.87	2.87	3.38	**2.71 ^a^**	6.78	10.24	15.24	**10.75 ^a^**
Mean	**1.69 ^C^**	**2.43 ^B^**	**3.34 ^A^**		**5.38 ^C^**	**9.24 ^B^**	**12.86 ^A^**	
ANOVA	**df**	***p*-value**		**LSD**		***p*-value**		**LSD**
Salinity (S)	2	< 0.001		0.02		< 0.001		0.15
Group (G)	3	< 0.001		0.05		< 0.001		0.14
S × G	6	< 0.001		0.10		< 0.001		0.27
Genotypes	**K^+^ (%)**				**K^+^/Na^+^ ratio**			
	**S1**	**S2**	**S3**	**Mean**	**S1**	**S2**	**S3**	**Mean**
Group A *n* = 1	19.97	16.50	11.32	**15.93 ^a^**	4.59	2.01	1.01	**2.54 ^a^**
Group B *n* = 6	19.77	15.09	10.00	**14.95 ^b^**	4.37	1.73	0.83	**2.31 ^b^**
Group C *n* = 8	18.30	13.41	9.41	**13.71 ^c^**	3.15	1.38	0.74	**1.75 ^c^**
Group D *n* = 3	17.27	12.36	8.29	**12.64 ^d^**	2.55	1.21	0.54	**1.43 ^d^**
Mean	**18.83 ^A^**	**14.34 ^B^**	**9.75 ^C^**		**3.66 ^A^**	**1.58 ^B^**	**0.78 ^C^**	
ANOVA	**df**	***p*-value**		**LSD**		***p*-value**		**LSD**
Salinity (S)	2	< 0.001		0.04		< 0.001		0.14
Group (G)	3	< 0.001		0.17		< 0.001		0.07
S × G	6	< 0.001		0.28		< 0.001		0.17

† Means in each column or row followed by the same lowercase or uppercase letter, respectively, are not significantly different at *p* ≤ 0.05 according to Fisher’s protected LSD test.

**Table 5 plants-09-01324-t005:** Impact of different salinity levels on plant height, spike length, number of spikes per square meter, number of grains per spike, 1000-grain weight, and grain yield of four genotype groups classified according to salinity tolerance under three salinity levels of S1 (5.25 dS m^−1^), S2 (8.35 dS m^−1^), and S3 (11.12 dS m^−1^).

Genotypes	Plant Height (cm)				Spike Length (cm)				Number of Spikes m^−2^			
	S1	S2	S3	Mean	S1	S2	S3	Mean	S1	S2	S3	Mean
Group A *n* = 1	92.50	88.67	74.00	**85.06 ^a^†**	12.31	10.85	9.89	**11.02 ^a^**	182.9	161.9	151.0	**162.2 ^a^**
Group B *n* = 6	87.54	76.36	67.91	**77.27 ^b^**	11.87	10.72	9.68	**10.76 ^b^**	173.6	154.3	133.3	**156.9 ^b^**
Group C *n* = 8	83.40	77.02	70.23	**76.88 ^bc^**	11.70	10.63	9.60	**10.64 ^b^**	144.9	130.0	117.8	**130.9 ^c^**
Group D *n* = 3	81.89	76.72	69.61	**76.07 ^c^**	11.13	10.28	9.13	**10.18 ^c^**	126.3	110.1	98.6	**111.7 ^d^**
Mean	**86.33 ^A^**	**79.69 ^B^**	**70.44 ^C^**		**11.75 ^A^**	**10.62 ^B^**	**9.58 ^C^**		**156.9 ^A^**	**139.1 ^B^**	**125.2 ^C^**	
ANOVA	**df**	***p*-value**		**LSD**		***p*-value**		**LSD**		***p*-value**		**LSD**
Salinity (S)	2	< 0.001		0.79		< 0.001		0.13		< 0.001		2.25
Group (G)	3	< 0.001		0.91		< 0.001		0.17		< 0.001		3.64
S × G	6	< 0.001		2.03		< 0.001		0.29		< 0.001		6.80
Genotypes	**Number of grains per spike**				**1000-grain weight (g)**				**Grain yield (kg ha^-1^)**			
	**S1**	**S2**	**S3**	**Mean**	**S1**	**S2**	**S3**	**Mean**	**S1**	**S2**	**S3**	**Mean**
Group A *n* = 1	59.17	54.00	44.17	**52.44 ^a^**	55.59	49.55	41.10	**48.75 ^a^**	4960.0	3973.0	2992.0	**3975.0 ^a^**
Group B *n* = 6	55.31	47.48	41.32	**48.03 ^b^**	53.16	47.55	40.78	**47.16 ^b^**	4732.3	3231.5	2217.8	**3393.9 ^b^**
Group C *n* = 8	51.60	44.65	40.04	**45.43 ^c^**	50.79	46.03	40.65	**45.82 ^c^**	3376.0	2687.9	2010.8	**2691.5 ^c^**
Group D *n* = 3	51.00	44.06	38.39	**44.48 ^d^**	48.66	44.01	39.00	**43.89 ^d^**	2708.3	2101.7	1458.7	**2089.6 ^d^**
Mean	**54.27 ^A^**	**47.54 ^B^**	**40.98 ^C^**		**52.05 ^A^**	**46.78 ^B^**	**40.38 ^C^**		**3944 ^A^**	**2999 ^B^**	**2170 ^C^**	
ANOVA	**df**	***p*-value**		**LSD**		***p*-value**		**LSD**		***p*-value**		**LSD**
Salinity (S)	2	< 0.001		0.52		< 0.001		0.40		< 0.001		74.58
Group (G)	3	< 0.001		0.77		< 0.001		0.62		< 0.001		124.3
S × G	6	0.003		1.33		< 0.001		1.03		< 0.001		203.4

† Means in each column or row followed by the same lowercase or uppercase letter, respectively, are not significantly different at *p* ≤ 0.05 according to LSD test.

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
