# Peer review of "Multidimensional Evaluation for Detecting Salt Tolerance of Bread Wheat Genotypes Under Actual Saline Field Growing Conditions"

_plants, 2020, doi:10.3390/plants9101324_

Round 1
Reviewer 1 Report
Suggestions and comments are, I expect, provided in attached PDF. Among these are the following specific comments:
L115. As explained on L146-147, it was not two experiments. Rather, you conducted a two-year field study. The mention of 'two field experiments' is confusing, because it suggests that chosen procedure(s) differed, such as cultivar, soil, salinity, or management differed between the two.
L137-139. When saline irrigation water is provided to plants, keeping the soil moisture near a saturated condition will not by itself avoid the occurrence of salinity stress.
L163-167. Please explain the number of plants measured per day. Measuring all 56 plants (n=3 x 3 x 18= 504 leaves) in two hours gives 0.20 seconds per leaf which is insufficient time to reach a 'steady-state' condition for leaf gas exchange. If so, the procedure is impractical and therefore, invalid.
L197. How many days from sowing to reach maturity? This is important since L160 indicates physiological parameters were measured at 55 d from sowing. Importantly, did maturity date differ among the 13 cultivars and 5 experimental lines?
L218-219. Please inform readers on the number of groups that were classified. Suggest revising state, "....salt tolerance, tolerant, moderately tolerant, moderately sensitive, and sensitive, based on their mean grain yield and YI across the three salinity levels." Please note, clarification is that classification is based, as I understand, on genotypic mean values across the three salinity levels.
L227-234. The information presented in Fig. 1 and Table 1 is redundant. I'm not familiar with dendogram. But unless it presents genotype pedigree (a genetic association), I suggest deleting Fig. 1.
L276. Revise to singular forms, salinity level, genotype, and genotype by salinity interaction.
L386-387. Suggest revising to '...the mean final grain yield and YI values across three salinity levels....'
L404-407. Good interpretation of results. I also note similarity of groups A and B groups for several of the measured parameters at the lowest salinity level (S1).
L466. Please note, the term 'irrigation water' is preferred over the term 'water irrigation', which is sometimes used in the text. Please revise accordingly.
L500-521. Suggest moving these statements to the Results section. Doing so would clarify the purpose of PCA and the meaning of its results in this salinity study, particularly for readers with limited knowledge of PCA.
L537-540. This statement is not supported by data in the tables, as significant difference (P<0.001) in all measured parameters was detected between each of the three irrigation water salinity levels. Did ANOVA produce a different result when the genotypes were not grouped according to the four salt-tolerance classes?
L542. Need to indicate the number of salt-tolerant genotypes. This is because the tables indicate one genotype was classified as salt tolerant. In short, the plural form 'genotypes, is incorrect unless this statement includes the six moderately-tolerant genotypes.

Author Response
Reviewer # 1
Response: We would like to deeply thank the reviewer for his appreciate time dedicated to our manuscript and his careful and thorough reading of the manuscript and giving constructive suggestions help for further improve quality of the manuscript. We believe that the manuscript is substantially improved after making the suggested revisions.
Suggestions and comments are, I expect, provided in attached PDF. Among these are the following specific comments:
- Line 57: What is the meaning of 'ambit'??
Response: The sentence has been revised.
- Line 57-60: No, plant ET will decrease as future rainfall decreases, but future temperature increase will increase the evaporative demand of atmosphere.
Response: That’s correct and the sentence is revised accordingly as suggested by the reviewer.
- Line 89: Based on title, I don't see how reference #37 relates to wheat cultivars.
Response: The reference #37 has been deleted (Allakhverdiev et al., 2000)
- Line 91: insert 'of' between 'Cultivation' and 'salt'
Response: Done
- Line 102: Revise to state, "....all of which occur simultaneously with...."
Response: Done
- Line 109: Explained well in Introduction section, so is probably not needed here.
Response: Has been deleted
- As explained on L146-147, it was not two experiments. Rather, you conducted a two-year field study. The mention of 'two field experiments' is confusing, because it suggests that chosen procedure(s) differed, such as cultivar, soil, salinity, or management differed between the two.
Response: That’s correct. It is one experiment in two seasons and the text has been modified accordingly.
- L137-139. When saline irrigation water is provided to plants, keeping the soil moisture near a saturated condition will not by itself avoid the occurrence of salinity stress.
Response: We agree with the reviewer that saturation will not avoid the occurrence of salinity and we rather meant lessen salt stress effect in this vulnerable stage. The sentence is modified to reveal the intended meaning.
- Line 126: Can you give the depth (mm or inch) of water applied? I calculate 19.5 inch irrigation water was applied during the season.
Response: The approximately applied amount of irrigation has been converted to mm.
- Line 141: add '(Table S3)' to end of this statement.
Response: Done
- Line 147: Revise to state, '...split-plot, randomized complete block design with three replicates in each growing season."
Response: Done
- Line 160: revise to '...and total chlorophyll concentration (Chl) was measured using ...' It is a concentration per leaf (mg/g FW), not a content per plant (g/plant or g/unit land area).
Response: Done
- Line 163-167. Please explain the number of plants measured per day. Measuring all 56 plants (n=3 x 3 x 18= 504 leaves) in two hours gives 0.20 seconds per leaf which is insufficient time to reach a 'steady-state' condition for leaf gas exchange. If so, the procedure is impractical and therefore, invalid.
Response: Many thanks for your comment. The time of measurements were performed from 9:30 AM to 12:30 PM over two consecutive days (about 6 hours in total). This means that each leaf takes approximately one minute. The sentence is clarified.
- How many days from sowing to reach maturity? This is important since L160 indicates physiological parameters were measured at 55 d from sowing. Importantly, did maturity date differ among the 13 cultivars and 5 experimental lines?
Response: Many thanks for your comment. About 110 days to maturity on average. Yes, there were slight differences in the maturity date (not more than 2-3 days between tested genotypes). The physiological parameters though were recorded before heading at 55 days and all cultivars were at the same vegetative growth stage.
- Line 204: Replace 'each' with 'individual' in line 204 and line 205.
Response: Revised as suggested by the reviewer
- Line 216: In my opinion, the variable names are singular (not plural). This also applies to interaction effect in the ANOVA. Please revise to singular form to be consistent with their presentation in Table 2. Also revise 'genotype' to 'genotype group' to be consistent with Tables 2, 3, and 4.
Response: Revised as suggested by the reviewer
- Lin 217: Please indicate if LSD test is 'protected' by a significant F test (P<0.05) in the ANOVA. I note Tables 2, 3, 4, and 5 indicate a protected LSD test for all parameters.
Response: Revised as suggested by the reviewer
- L218-219. Please inform readers on the number of groups that were classified. Suggest revising state, "....salt tolerance, tolerant, moderately tolerant, moderately sensitive, and sensitive, based on their YI across the three salinity levels." Please note, clarification is that classification is based, as I understand, on genotypic mean values across the three salinity levels.
Response: Revised as suggested by the reviewer
- Line 222: Suggest inserting '(PCA)' here and deleting the full name in remainder of the manuscript, line 310 for example.
Response: Revised as suggested by the reviewer
- L227-234. The information presented in Fig. 1 and Table 1 is redundant. I'm not familiar with dendogram. But unless it presents genotype pedigree (a genetic association), I suggest deleting Fig. 1.
Response: Fig. 1 has been deleted
- Revise to singular forms, salinity level, genotype, and genotype by salinity interaction.
Response: Revised as suggested by the reviewer
- Line 284: A more explicit descriptor is needed for 'cluster group' in Tables 2, 3, 4, and 5. Suggest using ''....of four genotype groups classified according to salinity tolerance under three salinity...."
Response: Revised as suggested by the reviewer
- Line 375: Subject-verb disagreement. Because 'tolerance' is the subject, need to revise 'were' to 'was'.
Response: Correct. Thank you.
- L386-387. Suggest revising to '... YI values across three salinity levels....'
Response: Revised as suggested by the reviewer
- Line 394: Revise 'a lack of chlorophyll content...' to 'a reductions in leaf chlorophyll concentration and photosynthetic efficiency'.
Response: Revised as suggested by the reviewer
- L404-407. Good interpretation of results. I also note similarity of groups A and B groups for several of the measured parameters at the lowest salinity level (S1).
Response: That’s very true. Thank you for pointing this out and for your constructive suggestions and comments.
- Line 439: Revise to state, ".., in association with decreased leaf K concentration and K/Na ratio (Table 4)".
Response: Revised as suggested by the reviewer
- Line 452: Revise 'located' with 'classified'.
Response: Revised as suggested by the reviewer
- Please note, the term 'irrigation water' is preferred over the term 'water irrigation', which is sometimes used in the text. Please revise accordingly.
Response: Many thanks for your comment. “irrigation water has been used in all text.
- Line 521: Suggest moving these statements to the Results section. Doing so would clarify the purpose of PCA and the meaning of its results in this salinity study, particularly for readers with limited knowledge of PCA.
Response: Revised as suggested by the reviewer
- Line 534: Suggest revising to state, "...had different genetic backgrounds and different responses to...."
Response: Revised as suggested by the reviewer
- L537-540. This statement is not supported by data in the tables, as significant difference (P<0.001) in all measured parameters was detected between each of the three irrigation water salinity levels. Did ANOVA produce a different result when the genotypes were not grouped according to the four salt-tolerance classes?
Response: Revised as suggested by the reviewer
- Need to indicate the number of salt-tolerant genotypes. This is because the tables indicate one genotype was classified as salt tolerant. In short, the plural form 'genotypes, is incorrect unless this statement includes the six moderately-tolerant genotypes.
Response: The moderately salt-tolerant genotypes are involved and the sentence has been rephrased

Reviewer 2 Report
The manuscript is focused on evaluation of salt tolerance of 18 bread wheat genotypes under 3 different actual saline field growing conditions.
The authors aimed to investigate the various physiological (alterations in total chlorophyll content; net photosynthesis rate; transpiration rate; stomatal conductance; membrane stability index; electrolyte leakage; relative water content; malondialdehyde; free proline, total soluble sugars, ascorbate, Cl-, Na+, K+ concentration and K+/Na+ ratio; activity of defense enzymes peroxidase, catalase, and superoxide dismutase) and agronomical (plant height; spike length; number of spikes; number of grains per spike; 1000-grain weight; grain yield; yield index) parameters of tested wheat genotypes in order to understand salt tolerance mechanisms and inter-relationship between agronomical and physiologic parameters under natural saline field conditions. The genotypes were divided into four groups ranging from salt-tolerant to salt-sensitive based on grain yield and yield index. More tolerant genotypes (groups A and B) were characterized with more efficient mechanism against salt stress which included higher contents of K+, photosynthetic pigments, and non-enzymatic osmolytes and higher gas exchange efficiency and antioxidant enzyme activities, accompanied by lower contents of Cl-, Na+, malondialdehyde and electrolyte leakage under high salinity level than salt-sensitive genotypes (groups C and D). Using principal component analysis the authors provided a comprehensive picture of the inter-relationships between agronomic, physiological and biochemical parameters, which can be used as the screening criteria for evaluation and improvement of the salt tolerance of wheat genotypes under natural saline field conditions.
The manuscript is well written and could be of interest of diverse specialists. It could be published after small modifications:
- Please specify from which leaf the material for determination of the biochemical parameters was collected.
- I think in Figure 1 (cluster denrdogram), group D and group C, it's better to be swapped.
- It is good the genotype codes in Table S3 to be numbered, for more convenient perception of the additional figures.
- In Figure S1 – bars of G12 and G13 for spike length are missing.
- In Figure S2 – legend symbol for 11.12 dS/m is missing
- Change MAD to MDA (line 540).
Author Response
Reviewer #2
The manuscript is focused on evaluation of salt tolerance of 18 bread wheat genotypes under 3 different actual saline field growing conditions.
The authors aimed to investigate the various physiological (alterations in total chlorophyll content; net photosynthesis rate; transpiration rate; stomatal conductance; membrane stability index; electrolyte leakage; relative water content; malondialdehyde; free proline, total soluble sugars, ascorbate, Cl-, Na+, K+ concentration and K+/Na+ ratio; activity of defense enzymes peroxidase, catalase, and superoxide dismutase) and agronomical (plant height; spike length; number of spikes; number of grains per spike; 1000-grain weight; grain yield; yield index) parameters of tested wheat genotypes in order to understand salt tolerance mechanisms and inter-relationship between agronomical and physiologic parameters under natural saline field conditions. The genotypes were divided into four groups ranging from salt-tolerant to salt-sensitive based on yield index. More tolerant genotypes (groups A and B) were characterized with more efficient mechanism against salt stress which included higher contents of K+, photosynthetic pigments, and non-enzymatic osmolytes and higher gas exchange efficiency and antioxidant enzyme activities, accompanied by lower contents of Cl-, Na+, malondialdehyde and electrolyte leakage under high salinity level than salt-sensitive genotypes (groups C and D). Using principal component analysis the authors provided a comprehensive picture of the inter-relationships between agronomic, physiological and biochemical parameters, which can be used as the screening criteria for evaluation and improvement of the salt tolerance of wheat genotypes under natural saline field conditions.
The manuscript is well written and could be of interest of diverse specialists. It could be published after small modifications:
Response: We would like to deeply thank the reviewer for his appreciate time dedicated to our manuscript and his careful and thorough reading of the manuscript and giving constructive suggestions help for further improve quality of the manuscript. We believe that the manuscript is substantially improved after making the suggested revisions.
- Please specify from which leaf the material for determination of the biochemical parameters was collected.
Response: Has been added
- I think in Figure 1 (cluster denrdogram), group D and group C, it's better to be swapped.
Response: Figure 1 has been deleted as suggested by reviewer 1 to avoid any redundancy with Table 1.
- It is good the genotype codes in Table S3 to be numbered, for more convenient perception of the additional figures.
Response: The genotype codes have been added to Table S3
- In Figure S1 – bars of G12 and G13 for spike length are missing.
Response: The bars have been added
- In Figure S2 – legend symbol for 11.12 dS/m is missing
Response: The legend has been added
- Change MAD to MDA (line 540).
Response: MAD has been changed to MDA
